# Effects of Golimumab and Ustekinumab on Circulating Dendritic Cell Migratory Capacity in Inflammatory Bowel Disease

**DOI:** 10.3390/biomedicines11102831

**Published:** 2023-10-18

**Authors:** Irene Soleto, Cristina Ramirez, Cristina Gómez, Montse Baldan-Martin, Macarena Orejudo, Jorge Mercado, María Chaparro, Javier P. Gisbert

**Affiliations:** Gastroenterology Unit, Centro de Investigación Biomédica en Red de Enfermedades Hepáticas y Digestivas (CIBERehd), Hospital Universitario de La Princesa, Instituto de Investigación Sanitaria Princesa (IIS-Princesa), Universidad Autónoma de Madrid (UAM), 28006 Madrid, Spain; irenesoletof@gmail.com (I.S.); ramicristina460@gmail.com (C.R.); crigom25@ucm.es (C.G.); mbaldanm@gmail.com (M.B.-M.); macaorejudo@gmail.com (M.O.); jorge.mercado.hlp@gmail.com (J.M.); mariachs2005@gmail.com (M.C.)

**Keywords:** dendritic cell, golimumab, ustekinumab, inflammatory bowel disease, ulcerative colitis, Crohn’s disease

## Abstract

Inflammatory bowel disease (IBD) is a chronic condition which includes ulcerative colitis (UC) and Crohn’s disease (CD), the origins of which are not yet fully understood. Both conditions involve an exacerbated immune response in the intestinal tract, leading to tissue inflammation. Dendritic cells (DCs) are antigen-presenting cells crucial for maintaining tolerance in the gastrointestinal mucosa. Previous research has indicated that DC recruitment to the intestinal mucosa is more pronounced in individuals with IBD, but the specific mechanisms governing this migration remain unclear. This study aimed to assess the expression of various homing markers and the migratory abilities of circulating DC subsets in response to intestinal chemotactic signals. Additionally, this study examined how golimumab and ustekinumab impact these characteristics in individuals with IBD compared to healthy controls. The findings revealed that a particular subset of DCs known as type 2 conventional DCs (cDC2) displayed a more pronounced migratory profile compared to other DC subsets. Furthermore, the study observed that golimumab and ustekinumab had varying effects on the migratory profile of cDC1 in individuals with CD and UC. While CCL2 did not exert a chemoattractant effect on DC subsets in this patient cohort, treatment with golimumab and ustekinumab enhanced their migratory capacity towards CCL2 and CCL25 while reducing their migration towards MadCam1. In conclusion, this study highlights that cDC2 exhibits a heightened migratory profile towards the gastrointestinal mucosa compared to other DC subsets. This finding could be explored further for the development of new diagnostic biomarkers or the identification of potential immunomodulatory targets in the context of IBD.

## 1. Introduction

Inflammatory bowel disease (IBD) is a persistent ailment characterized by an unclear underlying cause and currently lacks a cure. The primary forms of IBD encompass ulcerative colitis (UC) and Crohn’s disease (CD). UC primarily affects the colon and exhibits a continuous pattern of inflammation. In contrast, CD displays a sporadic inflammation pattern within various segments of the gastrointestinal (GI) tract, potentially leading to stenosis or fistulas. Both of these conditions follow a relapsing pattern, featuring periods of inactivity with no intestinal inflammation, referred to as “quiescent disease”, and active phases characterized by the manifestation of disease symptoms, known as “active disease [1]. While the exact cause of IBD remains elusive, it is believed to be a complex condition influenced by multiple factors. These include dysbiosis (an imbalance in the gut microbiota), genetic mutations that make individuals more susceptible to the disease, environmental factors, and an abnormal immune response against usually harmless antigens. Presently, IBD is a worldwide health concern, with a rising incidence particularly noted in Western nations. Over the period from 1990 to 2017, there has been a significant increase in the prevalence of IBD in numerous countries [2,3]. At local level, an incidence rate of 16 cases/100,000 people per year has been described for Spain [4]. In addition, the high cost of biological drugs entails a great economic burden for health systems. Within Europe, a direct cost of 4.6–5.6 billion per year is estimated [5].

Dendritic cells (DCs) are specialized antigen-presenting cells primarily found in mucosal surfaces and lymphoid tissues. Their primary function is to serve as a bridge between the innate and adaptive immune systems, facilitating the activation of naïve T cells and guiding their transformation into effector cells. Furthermore, DCs possess the capability to influence the delicate equilibrium between tolerance and immunity within the gastrointestinal (GI) mucosa. They achieve this by instructing naïve T cells to differentiate into either effector T cells, responsible for immune responses, or tolerogenic T cells, which help maintain tolerance and prevent excessive immune reactions in the GI tract [6]. In the human immune system, circulating dendritic cells (DCs) can be categorized into two main groups: plasmacytoid DCs (pDCs), identified by the presence of CD123 and breast cancer type 2 susceptibility protein (BDCA2) markers (CD123^+^ BDCA2^+^), and conventional DCs (cDCs), distinguished by the CD11c marker (CD11c^+^). Among the cDCs, there are further subdivisions into two types: type 1 (cDC1) characterized by the presence of CD141 and C-X-C motif chemokine receptor 1(CXCR1) markers (CD141^+^ CXCR1^+^), and type 2 (cDC2) identified by CD1c and signal regulatory protein α (SIRPα) markers (CD1c^+^ SIRPα^+^). Circulating cDC1 are specialized in a process known as cross-presentation, where they present endogenous antigens to CD8 T cells via MHC I molecules. This mechanism results in the generation of antigen-specific, cell-mediated cytotoxic lymphocyte responses, which are important for combating intracellular pathogens and cancerous cells. On the other hand, cDC2 primarily participate in classical antigen presentation mediated by MHC II molecules, which is essential for activating CD4 T cells and orchestrating adaptive immune responses [7,8].

In IBD, dysregulation of tolerance mechanisms for nutrients and commensals in the GI tract results in the development of effector T cells promoted by cDC [9]. It has been previously demonstrated that CD patients show a higher number of interleukin (IL)-23- and tumor necrosis factor alpha (TNFα) producing DC involved in disease pathogenesis [10]. In addition, elevated levels of toll-like receptors 2 and 4 and CD40 on DC from IBD patients indicates a high level of maturation [11,12]. The migration of conventional dendritic cells (cDCs) to the gastrointestinal (GI) tract is a well-established process, and it is primarily mediated by specific molecular interactions. One key player in this process is the integrin α4, which interacts with the mucosal vascular addressin cell adhesion Molecule 1 (MadCam1). This interaction helps guide cDCs to the mucosal lining of the GI tract. Additionally, the chemokine receptor 9 (CCR9) is involved in cDC migration to the GI tract, CCR9 binds to its ligand C-C motif chemokine ligand 25 (CCL25), which is secreted in the small bowel. This chemokine-receptor interaction plays a crucial role in directing cDCs to the small intestine. Furthermore, C-C chemokine receptor type 2 (CCR2) is another chemokine receptor involved in cDC migration, particularly in the context of the colon. It interacts with C-C motif chemokine ligand 20 (CCL20), which is secreted by colonic cells. This interaction facilitates the recruitment of cDCs to the colonic region of the GI tract. Together, these molecular interactions help orchestrate the precise migration of cDCs to different parts of the GI tract, where they play important roles in immune surveillance and maintaining immune balance [13,14,15].

Since there is currently no cure for IBD, the clinical approach is to induce clinical remission of the disease. Biologics have been demonstrated to be the most effective drugs in the maintenance of remission. However, only 30% of the patients achieve this therapeutic objective [16]. Furthermore, the medical community does not have criteria for selecting beneficial drugs for specific patients [17]. Following the failure of medical treatments, patients often resort to surgery, which can lead to both physical and social burdens. The precise mechanisms through which biological drugs exert their therapeutic effects in treating conditions like IBD are not yet fully understood. However, these drugs are believed to influence the cytokine environment within the gastrointestinal GI tract, thereby affecting the phenotype and function of GI DC and potentially altering the recruitment of circulating DC to the GI mucosa. For instance, biological drugs that target α4β7, work by inhibiting the migration of leukocytes toward the GI mucosa, which is a key process involved in the pathogenesis of IBD. It is worth noting that in IBD, there is an enhanced migration of cDC towards the GI tissue, and cDC homing markers have been found to correlate with disease phenotype. However, whether biological drugs targeting cytokines used in the treatment of IBD have the ability to modulate this migration of cDCs remains an area of current uncertainty and ongoing research [18,19]. Furthermore, a prior study conducted by our research group has already established that anti-TNFα and anti-α4β7 drugs can influence the migratory capabilities of various DC types in response to chemotactic signals within the gastrointestinal GI tract. Considering the pivotal role of DC in maintaining tolerance in the GI mucosa and the potential of certain biological drugs to alter DC migration, the objective of this study is to investigate the migratory capacity of both cDC and pDC towards chemoattractants in the GI mucosa. Additionally, the study aims to determine whether golimumab and ustekinumab, which are monoclonal antibodies targeting different cytokines, have the capacity to modulate this migratory process, both in healthy individuals and those affected by disease. This research seeks to shed light on how these biological drugs may impact the immune response within the GI tract, particularly in the context of health and disease.

## 2. Materials and Methods

### 2.1. Patients and Sample Collection

The study involved a total of 75 individuals, categorized into different groups, which included: 15 healthy controls (HC, 15 patients with ulcerative colitis (UC) exhibiting endoscopic activity (active UC), 15 patients with UC without endoscopic inflammation (quiescent UC), 15 patients with Crohn’s disease (CD) displaying endoscopic inflammation (active CD), 15 patients with CD without endoscopic inflammation (quiescent CD). The healthy control (HC) group consisted of individuals who were referred for various reasons, such as changes in bowel transit, colorectal cancer screening, or rectal bleeding. Importantly, all individuals in the HC group exhibited both macroscopically and histologically normal mucosal tissue and did not have any known inflammatory, autoimmune, or malignancy diseases. This study (JPG-VED-2016-01) received ethical approval from the local ethics committee at La Princesa Hospital in Madrid, Spain. Furthermore, all participating patients provided written informed consent for both their involvement in the study and the collection of biological samples.

Each individual contributed 20 mL of blood, which was collected in a single visit and immediately processed in the laboratory for subsequent analysis. For additional demographic information about the patients, please refer to Appendix A.

### 2.2. Blood Processing

Peripheral blood mononuclear cells (PBMCs) were isolated through a centrifugation process using Ficoll-Paque PLUS (Amersham Biosciences, Amersham, UK). Subsequently, these PBMCs were subjected to two washes with a complete medium, which comprised RPMI 1640 (Sigma-Aldrich, St. Louis, MO, USA), 100 μg/mL penicillin/streptomycin, 2 mM L-glutamine, 50 ug/mL gentamicin (Sigma-Aldrich), and 10% fetal bovine serum (TCS cellworks). The PBMCs were then frozen and preserved at −80 °C until required for further analysis. 

Upon retrieval, the cells were thawed by immersing them in a 37 °C water bath. Following thawing, the PBMCs underwent centrifugation at 1500× *g* rpm for 5 min, after which the supernatants were carefully removed. Subsequently, the PBMCs were subjected to staining while suspended in PBS containing 1 mM EDTA and 0.02% sodium azide (referred to as FACS buffer). This staining process involved the use of fluorochrome-conjugated antibodies, as further detailed below, or the cells were cultured for subsequent experiments.

### 2.3. Antibody Labeling

Peripheral blood mononuclear cells (PBMCs) underwent staining with monoclonal antibodies and were subsequently characterized using flow cytometry. To ensure accurate analysis, a live/dead fixable near-IR dead cell stain kit from molecular probes was employed, allowing for the exclusion of dead cells from the analysis. Appendix A provides details regarding the specificity, clone, fluorochrome, and sources of the antibodies used in this study.

The staining process occurred in FACS buffer on ice and in the dark, with cells being incubated for 20 min to minimize nonspecific binding. Circulating pDC, cDC1, and cDC2 were identified within the population of viable, singular leukocytes. These subsets were further examined for the expression of various homing markers (C-C chemokine receptor type 2,5,6 and 9-CCR2, CCR5, CCR6, CCR9- and β7) that are associated with their migration towards the GI tract. This flow cytometry-based approach allowed for a comprehensive characterization of these immune cell subsets and their migratory profiles.

### 2.4. PBMC Culture

In the study, PBMCs obtained from both HC and individuals with IBD were subjected to a culture process. Specifically, PBMCs were cultured in complete medium at a concentration of 1 million PBMCs per 1 mL. This culture was conducted at a temperature of 37 °C for a duration of 18 h.

During the culture period, two biological drugs, golimumab at 10 μg/mL, and ustekinumab at 10 μg/mL were added to the culture medium. For comparison and as internal and negative controls, a set of paired PBMCs were cultured in complete medium but in the absence of any drug. These control PBMCs were used to establish a baseline against which the effects of golimumab and ustekinumab could be evaluated.

### 2.5. Transwell Migration Experiments

This study evaluated the migratory capacity of circulating DC subsets from different study groups, including HC and individuals with IBD, both with and without prior conditioning. To assess this migration, 3 μm pore polycarbonate membrane culture inserts (transwell inserts) were employed. Migration experiments were conducted using the following conditions:

Basal control: migration towards complete culture medium (no specific chemokine stimulus). CCL2-stimulated migration: migration towards medium supplemented with 100 ng/mL of CCL2 (a ligand for CCR2, a chemokine receptor).CCL25-stimulated migration: migration towards medium supplemented with 500 ng/mL of CCL25 (a ligand for CCR9, another chemokine receptor).

MadCAM1-stimulated migration: Migration towards medium supplemented with 1 μg/mL of MadCAM1 (a ligand for β7 integrin).

The migration assays were conducted over a 4-h period. Subsequently, the cells that had migrated were harvested and analyzed using flow cytometry. This approach allowed for the assessment of the migratory responses of DC subsets from different study groups under various chemotactic stimuli, providing valuable insights into their migratory behavior in the context of health and inflammatory bowel disease.

### 2.6. Flow Cytometry and Data Analysis

In this study, cell data were acquired using two different flow cytometers:

For characterizing dendritic cells (DCs), an LSR-Fortessa flow cytometer from BD Biosciences was used and for migration assays, a BD Canto II flow cytometer, also from BD Biosciences, was employed.

The acquired data from both flow cytometers were subsequently analyzed using FlowJo software 10.1. During the analysis process, all cells were confined to the singlet viable fraction, ensuring that only individual, viable cells were considered for analysis. For setting positive and negative gates to distinguish specific populations of interest, the FMO (fluorescence minus one) method was used. This method involves creating control samples for each fluorochrome in which one of the fluorescent markers is omitted, allowing for the accurate determination of positive and negative signals in the experimental samples. This approach helps ensure the reliability and consistency of the flow cytometry data analysis.

### 2.7. Statistical Analysis

In this study, the individual characteristics of patients were reported, including age, sex, and the treatments they received for inflammatory bowel disease (IBD). Categorical variables were presented as the sample size (percentage), while quantitative variables were expressed as mean ± standard deviation (SD). Data analysis was conducted using GraphPad Prism 6.01 software (San Diego, CA, USA). To examine differences between quantitative variables, statistical tests such as one-way or two-way analysis of variance (ANOVA) were utilized, depending on the specific study design (with or without repeated measures). Following the ANOVA, Tukey post-hoc correction was applied to assess specific group differences, as indicated in each figure legend. The significance threshold was set at *p* < 0.05 for all statistical analyses, indicating that results with *p*-values below this threshold were considered statistically significant. This approach allowed for the rigorous examination of data and the identification of meaningful differences between study groups and variables.

## 3. Results

### 3.1. Differential Homing Marker Profile in DC from HC and IBD Patients

In this study, a total of 75 individuals were enrolled for analysis. The statistical analysis revealed significant differences in age and sex when comparing HC to individuals with IBD. However, within the various sub-groups of IBD patients, there were no statistically significant differences observed in terms of age or sex, as indicated in Table 1.

In this study, human circulating dendritic cell (DC) subsets were identified within a distinct population of viable cells characterized by the absence of CD19, HLA-DR, and CD14 markers, denoted as CD19^−^ HLA-DR^+^ CD14^−^. These DCs were further categorized into two main subsets: plasmacytoid DCs (pDCs), identified by the expression of CD123 (CD123^+^), and conventional DC (cDC), distinguished by the presence of CD11c (CD11c^+^). Within the cDC subset, further subtyping was carried out, classifying them into cDC1 (CD141^+^) and cDC2 (CD1c^+^) based on the expression of CD141 and CD1c markers (Figure 1A).

To characterize the migratory behavior of all DC subsets from HC, their expression levels of specific markers, including β7, CCR2, CCR5, CCR6, and CCR9, were analyzed. Notably, the study revealed that cDC2 exhibited higher levels of expression for β7 and CCR6 when compared to pDCs and cDC1. Additionally, cDC2 demonstrated elevated CCR2 expression compared to cDC1 (Figure 1B).

After characterizing the homing-marker profiles of different dendritic cell (DC) subsets during homeostasis, this study investigated changes in their expression levels during various stages of inflammatory bowel disease (IBD) inflammation, including active IBD patients (ulcerative colitis—aUC and Crohn’s disease—aCD) as well as quiescent patients (ulcerative colitis—qUC and Crohn’s disease—qCD). It was observed that pDC and cDC1 exhibited similar homing-marker profiles across all groups. There were no significant differences in the percentages of cells expressing integrin β7 and CCR5 in any of these DC subsets across the studied groups. In contrast, cDC2 in aUC patients displayed lower expression levels of both CCR2 and CCR6 compared to those in quiescent ulcerative colitis (qUC) patients. Furthermore, the expression of CCR6 in cDC2 from aUC patients was lower than that in HC. The expression of CCR9 in cDC2 showed increased levels in both quiescent patient groups (qUC and qCD) compared to both HC and aUC patients. However, this increase was statistically significant for qUC patients (Figure 2).

### 3.2. DC Migration Capacity toward GI Chemoattractants 

After the study of migratory profiles of DC subsets in homeostasis and different inflammatory stages of IBD, we focused on assaying whether the differential expression of these markers represents functional changes. We carried out a transwell migration assay toward the ligands of CCR2, CCR9, and integrin β7 (CCL2, CCL25, and MadCam 1, respectively). The results were normalized to the spontaneous migration toward non-supplemented cultured medium (basal migratory capacity). Unlike in our previous work, none of the chemoattractants induced an increase of migratory capacity in any DC subset across groups, compared to the basal migratory capacity (Figure 3).

### 3.3. Golimumab and Ustekinumab Modify the DC Migratory Profile of IBD Patients

We then explored whether the altered expression of homing markers, particularly in cDC2 subsets, observed in patients with IBD could be influenced or modulated by biological drugs. To investigate this, they conducted experiments where PBMCs from HC and IBD patients at different stages of disease activity were incubated with two different biological drugs: anti-TNFα (golimumab) and anti-IL12/23 (ustekinumab). We found that neither golimumab nor ustekinumab had any significant impact on the expression of surface molecules in pDC or cDC2 across any of the study groups. In other words, the expression profiles of these surface molecules remained largely unchanged after treatment with these drugs. In contrast, in the cDC1 subset, both golimumab and ustekinumab had specific effects on the expression of chemokine receptors. Golimumab led to an increase in CCR6 expression in qCD patients, while ustekinumab increased CCR9 expression in aUC patients (Figure 4).

### 3.4. Golimumab and Ustekinumab Modify the Migratory Capacity of cDC

In the final part of the study, we investigated whether golimumab and ustekinumab, which were found to modify dendritic cell (DC) homing-marker expression, could also impact the migratory capacity of DCs towards specific chemokines, including CCL2, CCL25, and MadCam1.

As previously shown in Figure 3, neither DC from HC nor those from IBD patients migrated in response to the different chemoattractants analyzed compared to their basal migratory capacity. However, treatment with ustekinumab of cDC1 from aUC patients significantly increased their migration toward CCL2 compared with their basal migration, but also compared with that of untreated cells (Figure 5). In contrast, cDC1 from aCD patients treated with ustekinumab showed lower migration toward MadCam1 compared with untreated cells.

Regarding cDC2, we found that conditioning with both golimumab and ustekinumab increased migration toward CCL2 in cells from HC. Additionally, golimumab treatment increased migration toward CCL25 compared with untreated cells and basal migration condition in cDC2 from HC but not in those from IBD patients. Lastly, pDC migration was not modulated by golimumab or ustekinumab (Figure 5).

## 4. Discussion

We herein report how golimumab and ustekinumab increase the migratory capacity of cDC from aUC patients and HC toward CCL2 and CCL25 and decrease the migratory capacity of cDC1 from aCD patients toward MadCam1. With the current work, the knowledge on how these biological drugs act on the systemic immune response is increased. In addition, we provide knowledge on how to modify the DC migratory profile, and how this affects the recruitment of these cells to the GI mucosa. Our results demonstrate that the percentage of cells expressing homing receptors such us β7, CCR2, and CCR6 is higher in cDC2 than in other DC subsets. Similarly, we previously published that the cDC2 migratory profile is more active than that of other DC subsets [20]. Therefore, the findings of this new cohort of patients strongly support the role of cDC2 as a potential target candidate to design new treatments to improve the management of IBD and other inflammatory diseases, where an aberrant behavior of these cells has been reported, such as arthritis [21], psoriasis [22] or lupus nephritis [23]. In agreement with the previous results of our group, the percentage of CCR6+ cDC2 was lower in aUC patients compared with HC and qUC subjects [20]. Nevertheless, it has been previously reported that the CCL20/CCR6 axis is transcriptionally upregulated in colonic samples from aUC patients but not in qUC [24]. Moreover, this axis has been used as a drug target demonstrating promising results in mouse models [25]. CCL20 is secreted by epithelial cells at certain inflammation stages, increasing the recruitment of immature DC in a CCR6-dependent manner [26]. However, the maturation stages of the migrated cells in this study were not evaluated, so deeper studies are necessary to clarify the discrepancy with our results. Unlike CCR6 expression, the percentage of CCR9^+^ cDC2 cells is increased in quiescent patients in both UC and CD, and is statistically significant in the case of UC. The findings presented in this study align with previous research conducted by our group, which also indicated that CCR9 plays a protective role in IBD [20,27]. This consistency in results strengthens the evidence for the significance of CCR9 in IBD pathogenesis and suggests its potential as a diagnostic biomarker. Further investigations should be undertaken to assess CCR9 expression specifically on cDC2. If this association is confirmed, it could have practical implications in clinical settings. CCR9 expression on cDC2 may serve as a valuable diagnostic biomarker for IBD. This could streamline the diagnostic process and potentially reduce the need for invasive procedures like colonoscopy. Multiparameter flow cytometry, a well-established tool in the diagnosis of various diseases, may be a useful method for assessing CCR9 expression in clinical practice [28]. Although the CCR9-CCL25 axis has been proposed as a drug target in the treatment of IBD, the CCR9 antagonist Vercirnon, which showed encouraging results in early phase clinical trials, failed to meet the primary endpoint in a subsequent phase 3 trial [29]. The mention of a clinical trial involving an anti-MadCam1 drug and its impact on CCR9 expression in active Crohn’s disease (aCD) patients is significant. It suggests that CCR9 serves as a relevant pharmacodynamic biomarker in the context of this specific treatment. The increase in CCR9 expression observed following treatment with the anti-MadCam1 drug in aCD patients indicates that CCR9 may play a role in the mechanism of action of this therapeutic intervention. This finding highlights the potential utility of CCR9 as a biomarker to monitor treatment response and evaluate the effectiveness of anti-MadCam1 therapies in aCD [30]. Interestingly, integrin β7 and CCR5 were not modulated in any DC subset in any study group, maybe because their expression is predominant in other cell types like B or T cells. 

We then focused on the migrating capacities of these cell subsets, given that some of their homing markers are modulated during IBD. In a previous study from our group, we demonstrated that CCL2 increased the migration of DC [20]. Unexpectedly, what we saw in this case is that none of the chemoattractants analyzed increased the migration of DC compared with our basal conditions. The main difference between both studies is the expression of CCR2, which was higher in the cohort of our previous work. This could explain the lack of an effect of CCL2 on the migration of the different DC populations in the current study. 

Analyzing the effect of golimumab and ustekinumab on the expression of migratory markers on DC subsets, only cDC1 cells showed modulation of their surface markers by these drugs. Both golimumab and ustekinumab increased the percentage of CCR6^+^ cDC1 only in qCD patients. In agreement with these data, it has been reported that the percentage of intradermal CCR6^+^ CD8 T cells is increased in psoriasis resolved by treatment with biologics (anti-TNF and ustekinumab) [31]. Although CCR9^+^ cDC1 percentage was also increased in aUC patients treated with golimumab and ustekinumab, potentially increasing their migratory capacities, the maturation and activation state of this DC subset was unknown. Accordingly, future studies are essential to clarify this question. 

Regarding DC migratory capacities, we found that cDC1 and cDC2 treated with golimumab and ustekinumab increased their migration capacity toward CCL2 and in the case of cDC2 also toward CCL25, which indicates that anti-TNF and anti-IL12/23 have the capacity to increase the migration of cDC toward the GI mucosa. Similar results with other anti-TNF drugs have been published previously [20]. As we mentioned previously, we did not study the activation or maturation state of these cells. If this increment in the migratory capacities of cDC were only exerted on tolerogenic cells, it would represent a new mechanism of action of anti-TNF and ustekinumab. To confirm this possibility, future studies should be carried out to analyze the characteristics of migrated cells. In this regard, it has been reported that ustekinumab decreases the transcriptional expression of CCL2 in the ilium of CD patients [32], thus reinforcing the notion of additional mechanisms of action for ustekinumab besides the inhibition of IL12 and IL23. Further studies are needed to describe these mechanisms. 

In summary, in the current study we found that cDC2 express more homing markers than other DC subsets, which are also modulated during IBD progression, thereby suggesting that this DC subset cell might be a potential biomarker and drug target. Nevertheless, further studies are needed to validate these results. Moreover, this study demonstrates that the expression of CCR9 on cDC2 plays a protective role in IBD and thus could be considered a possible diagnostic biomarker. If this is corroborated, it could be useful to avoid unnecessary colonoscopies. In addition, both golimumab and ustekinumab increase the migratory capacities of cDC towards CCL2 and CCL25, suggesting an additional mechanism of action for these biological drugs based on the control of DC migration.

## Figures and Tables

**Figure 1 biomedicines-11-02831-f001:**
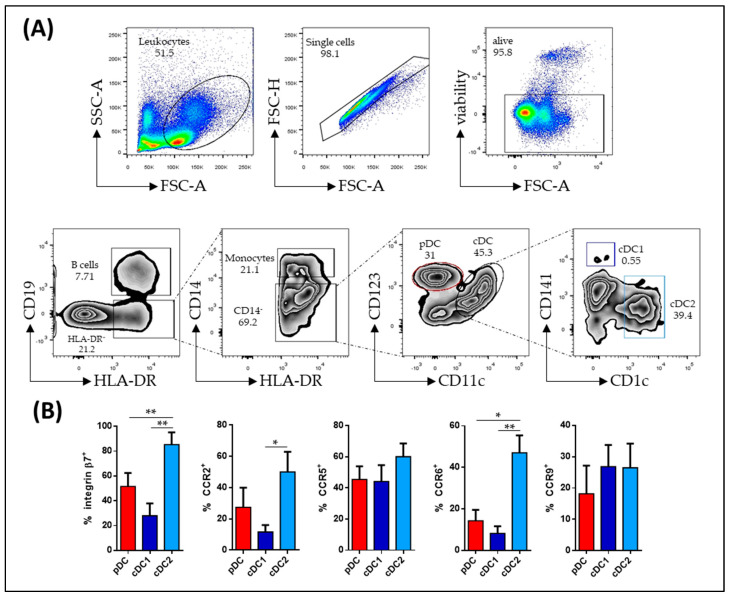
Phenotypic characterization of circulating dendritic cell subsets. (**A**) Dendritic cell (DC) populations were identified using flow cytometry within singlet viable peripheral blood mononuclear cells. These DCs were characterized as HLA-DR^+^ CD19^−^ CD14^−^ cells. Among the CD14-negative cells, two main subsets were identified: plasmacytoid dendritic cells (pDCs), characterized as CD123^+^ CD11c^−^, and myeloid dendritic cells (mDCs), characterized as CD123^+^ CD11c^+^. Further subtyping of mDCs was based on the expression of CD141 (cDC1) and CD1c (cDC2 (**B**) The phenotype of different DC subsets from healthy controls was determined by analyzing the expression of specific markers, including integrin β7, CCR2, CCR5, CCR6, and CCR9. The results are presented as the percentage of positive cells (%) and are represented as the mean ± SEM, with a sample size (*n*) ranging from 10 to 15 individuals. To assess the differences in basal marker expression among pDCs, cDC1, and cDC2, a statistical analysis was conducted using one-way ANOVA with repeated measures, followed by Tukey correction. Significance was determined with *p*-values less than 0.05 considered statistically significant (* *p* < 0.05, ** *p* < 0.01).

**Figure 2 biomedicines-11-02831-f002:**
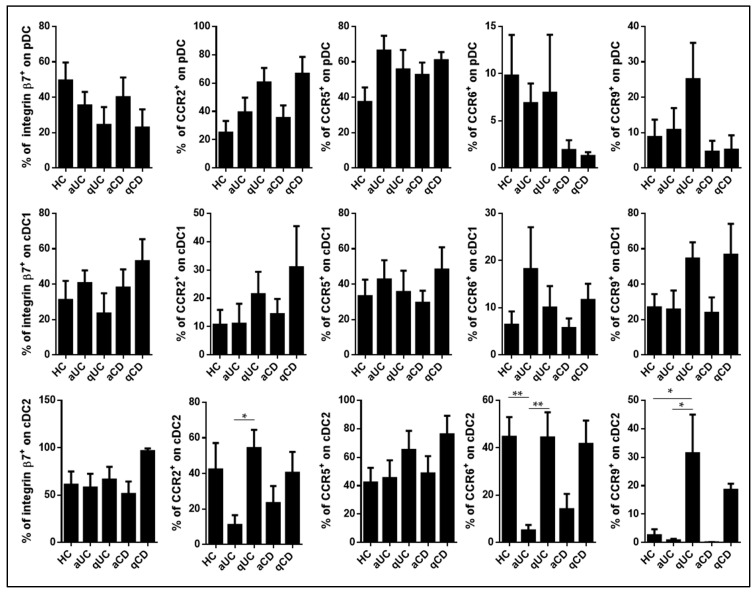
Integrin β7, CCR2, CCR5, CCR6, and CCR9 expression on circulating dendritic cells from controls and patients with inflammatory bowel disease. Dendritic cell (DC) subsets were identified using the same method as shown in Figure 1. The study included individuals from various groups, including healthy controls (HCs), patients with active ulcerative colitis (aUC), quiescent ulcerative colitis (qUC), active Crohn’s disease (aCD), and quiescent Crohn’s disease (qCD). The expression of specific markers, including integrin β7, CCR2, CCR5, CCR6, and CCR9, was assessed within each DC subset for each study group. The results are presented as the percentage of positive cells (%) and are depicted as the mean ± SEM, with a sample size (*n*) ranging from 10 to 15 individuals in each group. To compare the expression levels of integrin β7, CCR2, CCR5, CCR6, and CCR9 on plasmacytoid dendritic cells (pDC), conventional dendritic cells type 1 (cDC1), and conventional dendritic cells type 2 (cDC2) among the different study groups, a statistical analysis was performed using one-way ANOVA with Tukey correction. In this analysis, *p*-values less than 0.05 were considered statistically significant (* *p* < 0.05, ** *p* < 0.01).

**Figure 3 biomedicines-11-02831-f003:**
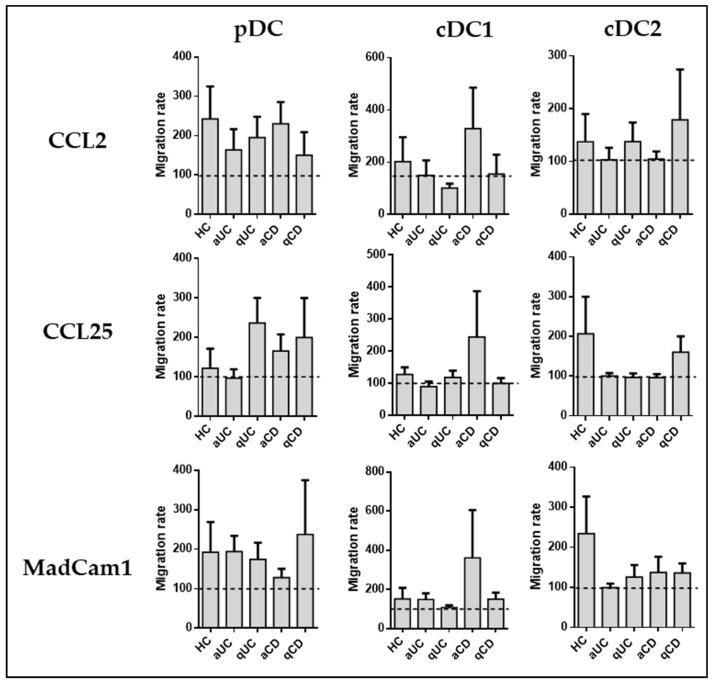
Dendritic cell subset migration is not altered in patients with inflammatory bowel disease. Peripheral blood mononuclear cells (PBMCs) obtained from various groups, including healthy controls (HCs), patients with active ulcerative colitis (aUC), quiescent ulcerative colitis (qUC), active Crohn’s disease (aCD), and quiescent Crohn’s disease (qCD), were subjected to migration assays using gut-homing chemoattractants, specifically CCL2, CCL25, and MadCam1. The migration assays were performed with dendritic cell (DC) subsets, which were identified as described in Figure 1. The results of these migration assays were normalized to the spontaneous migration of the cells toward non-supplemented culture medium (basal condition, represented by the dotted line). The findings are presented as the mean ± SEM, with a sample size (*n*) of 13 individuals in each group. Statistical analysis was conducted using one-way ANOVA with Tukey correction to compare the migration of DC subsets towards the basal condition.

**Figure 4 biomedicines-11-02831-f004:**
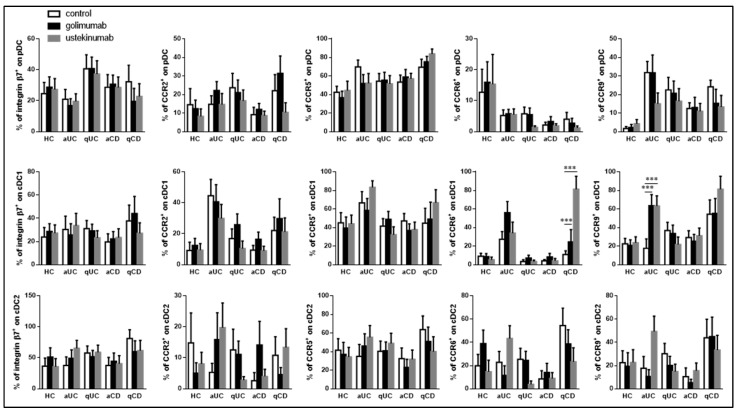
Ustekinumab and golimumab upregulate CCR6 and CCR9 expression on cDC. Dendritic cell (DC) subsets were identified using the same method as shown in Figure 1. The study included individuals from various groups, including healthy controls (HCs), patients with active ulcerative colitis (aUC), quiescent ulcerative colitis (qUC), active Crohn’s disease (aCD), and quiescent Crohn’s disease (qCD). The expression of specific markers, including integrin β7, CCR2, CCR5, CCR6, and CCR9, was assessed within each DC subset. The DC subsets were further studied after conditioning with two biological drugs, golimumab (black bars) and ustekinumab (grey bars). The results are presented in comparison to untreated controls (white bars). The percentages of positive cells expressing these markers are depicted as the mean ± SEM, with a sample size (*n*) ranging from 10 to 15 individuals in each group. To compare the expression levels of integrin β7, CCR2, CCR5, CCR6, and CCR9 on plasmacytoid dendritic cells (pDC), conventional dendritic cells type 2 (cDC2), and conventional dendritic cells type 1 (cDC1) between the different groups and conditions, a statistical analysis was performed using one-way ANOVA with subsequent Tukey post-hoc correction. Statistically significant differences were denoted with asterisks as follows: *** for *p* < 0.001.

**Figure 5 biomedicines-11-02831-f005:**
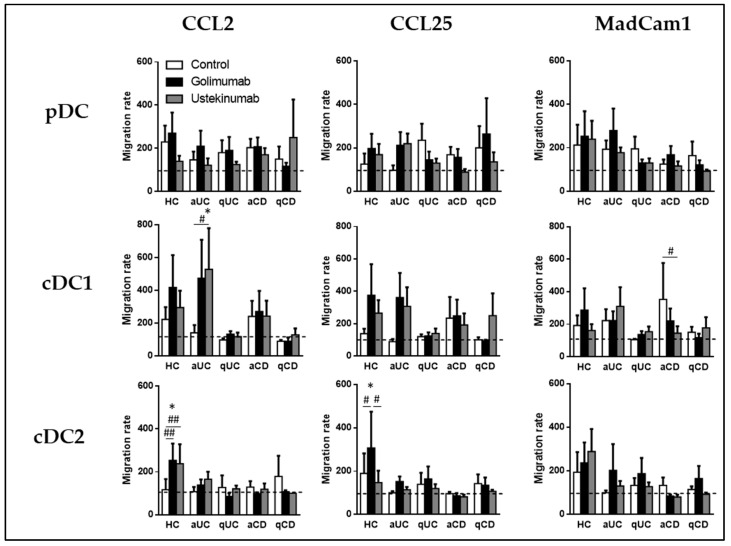
Golimumab and ustekinumab modulate dendritic cell subsets’ migratory capacity in healthy controls and IBD patients. Peripheral blood mononuclear cells (PBMCs) from different groups, including healthy controls (HCs), patients with active ulcerative colitis (aUC), quiescent ulcerative colitis (qUC), active Crohn’s disease (aCD), and quiescent Crohn’s disease (qCD), were subjected to migration assays. These assays were conducted to assess the migratory capacity of dendritic cell (DC) subsets, including plasmacytoid dendritic cells (pDC), conventional dendritic cells type 2 (cDC2), and conventional dendritic cells type 1 (cDC1), in response to gut-homing chemoattractants, namely CCL2, CCL25, and MadCam1. The cells were conditioned with either golimumab (black columns) or ustekinumab (grey columns), and their migration was compared to migration toward non-supplemented complete medium (white bar). The results were further normalized to the migration observed under basal conditions (spontaneous migration, represented by the dotted line). The findings are presented as the mean ± SEM, with a sample size (*n*) ranging from 10 to 15 individuals in each group. Statistical analysis was performed using one-way ANOVA with subsequent post-hoc Tukey correction. This analysis was conducted to determine differences in migration within each subset and condition compared with basal or spontaneous migration (denoted as *) and to compare migration within each patient group between each culture condition (denoted as #). Statistically significant differences are indicated as follows: * for *p* < 0.05, # for *p* < 0.05, and ## for *p* < 0.01.

**Table 1 biomedicines-11-02831-t001:** The study cohort (*n* = 75) consisted of individuals categorized into different groups.

Variable	HC	aUC	qUC	aCD	qCD
Age (years) mean ± SD	32.56 ± 6.12	46.73 ± 14.84	50.66 ± 11.83	44.86 ± 16.30	53.2 ± 15.15
Sex (Male), *n* (%)	3 (20%)	8 (53.3%)	10 (66.6%)	7 (46.6%)	7 (46.6%)

Including healthy controls (HCs), patients with active ulcerative colitis (aUC), those with quiescent ulcerative colitis (qUC), patients with active Crohn’s disease (aCD), and those with quiescent Crohn’s disease (qCD). The characteristics of this cohort were reported as follows: continuous variables were presented as mean ± SD, and categorical variables were expressed as numbers (percentage). To compare the ages among the various groups, a one-way ANOVA with repeated measures and Tukey correction was employed. Differences in sex distribution among the groups were assessed using a χ^2^ test. Patient categories were depicted as the sample size (percentage) and age (mean ± SD).

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
