# Peer review of "Effects of Golimumab and Ustekinumab on Circulating Dendritic Cell Migratory Capacity in Inflammatory Bowel Disease"

_biomedicines, 2023, doi:10.3390/biomedicines11102831_

Round 1
Reviewer 1 Report
In this study, authors investigated the roles of golimumab and ustekinumab in circulating dendritic cell migratory capacity in inflammatory bowel disease. They found that type 2 conventional DC (cDC2) had a more prominent migratory profile than other DC subsets. In addition, golimumab and ustekinumab modified the migratory profile of cDC1 differentially in CD and UC. When DCs were treated with golimumab and ustekinumab their migratory capacity toward CCL2 and CCL25 was increased while that toward MadCam1 was decreased. They concluded that cDC2 present a more migratory profile toward the gastrointestinal mucosa than other DC subsets, which should be further studied to develop new diagnostic biomarkers or to identify immunomodulatory targets in IBD. Generally, this study is interesting. They expanded the knowledge regarding IBD and provided better understanding of the characteristics of IBD. Here are some comments from this reviewer:
1. Please provide the approved ethical protocol number of human study.
2. How do you enrich DCs for the migration assays?
3. Do circulating DCs and intestinal resident DCs have the same features? How about the intestinal resident DCs?
4. Do Golimumab and ustekinumab modify the migratory capacity of DC in IBD animal models?
Author Response
Reviewer #1
In this study, authors investigated the roles of golimumab and ustekinumab in circulating dendritic cell migratory capacity in inflammatory bowel disease. They found that type 2 conventional DC (cDC2) had a more prominent migratory profile than other DC subsets. In addition, golimumab and ustekinumab modified the migratory profile of cDC1 differentially in CD and UC. When DCs were treated with golimumab and ustekinumab their migratory capacity toward CCL2 and CCL25 was increased while that toward MadCam1 was decreased. They concluded that cDC2 present a more migratory profile toward the gastrointestinal mucosa than other DC subsets, which should be further studied to develop new diagnostic biomarkers or to identify immunomodulatory targets in IBD. Generally, this study is interesting. They expanded the knowledge regarding IBD and provided better understanding of the characteristics of IBD. Here are some comments from this reviewer:
- Please provide the approved ethical protocol number of human study.
We thank to the reviewer for his comment , we include the ethical protocol number (JPG-VED-2016-01) in the manuscript.
Line 129: malignancy diseases. This study (JPG-VED-2016-01) received ethical approval
- How do you enrich DCs for the migration assays?
We thank to the reviewer for his comment, we chose not to isolate dendritic cells (DC) from patients as we aimed to create a more "physiological model" by enabling them to migrate within the peripheral blood mononuclear cell (PBMC) setting.
- Do circulating DCs and intestinal resident DCs have the same features? How about the intestinal resident DCs?
As suggested by the reviewer, resident dendritic cells (DC) exhibit distinct characteristics compared to circulating DC. DCs are renowned for being the most potent antigen-presenting cells, possessing the unique ability to initiate primary immune responses by activating naive T cells. Furthermore, they play a crucial role in determining the outcome of immune responses, whether they tend towards tolerance or a proinflammatory state. DC precursors initially migrate from the bone marrow to the gastrointestinal (GI) mucosa, where they assume the role of sentinels and immune system sensors. DCs actively sample their surroundings, demonstrating exceptional proficiency in capturing and processing antigens.
In humans, dendritic cells from the gastrointestinal mucosa (GI-DC) typically display a tolerogenic phenotype under resting conditions, characterized by low expression of Toll-like receptors (TLRs), which limits their capacity to recognize bacterial antigens. These human GI-DCs can be categorized into CD103-SIRPα+, CD103+SIRPα+, and CD103+SIRPα- subsets. Specifically, CD103+SIRPα- DCs originate from cDC1, while cDC2 cells enter the mucosa initially as CD103-SIRPα+ cells through a CCR2-dependent process. Subsequently, they transform into CD103+SIRPα+ DCs following exposure to mucosal tolerogenic factors such as TGF-β.It's important to note that the phenotype and function of DCs are profoundly influenced by the local microenvironment. Thus, DCs acquire a tolerogenic "gut-like" profile in the presence of a colonic microenvironment, while the pro-inflammatory cytokine milieu observed in inflammatory bowel disease (IBD) disrupts this balance.
Given the dependency of DC phenotype on the pro-inflammatory profile, the authors decided to investigate whether biological drugs designed based on alternative mechanisms could modulate the migratory capacities of circulating DCs toward the GI mucosa.Principio del formulario
- Ortega Moreno L, Fernández-Tomé S, Chaparro M, Marin AC, Mora-Gutiérrez I, Santander C, Baldan-Martin M, Gisbert JP, Bernardo D. Profiling of Human Circulating Dendritic Cells and Monocyte Subsets Discriminates Between Type and Mucosal Status in Patients With Inflammatory Bowel Disease. Inflamm Bowel Dis. 2021 Jan 19;27(2):268-274.
- Hart AL, Al-Hassi HO, Rigby RJ, Bell SJ, Emmanuel AV, Knight SC, Kamm MA, Stagg AJ. Characteristics of intestinal dendritic cells in inflammatory bowel diseases. Gastroenterology. 2005 Jul;129(1):50-65.
- Do Golimumab and ustekinumab modify the migratory capacity of DC in IBD animal models?
We appreciate the reviewer's pertinent question. To the best of our knowledge, there have been no prior studies investigating the impact of Golimumab and Ustekinumab on this specific aspect. This article represents a unique endeavor in examining the potential effects of these drugs.
It's worth noting that the only authorized biological drug currently involved in the modulation of leukocyte trafficking is Ustekinumab, as it targets an integrin that plays a role in these mechanisms. This sets it apart as a noteworthy subject of investigation in this study.
- Schulze LL, Becker E, Dedden M, Liu LJ, van Passen C, Mohamed Abdou M, Müller TM, Wiendl M, Ullrich KA, Atreya I, Leppkes M, Ekici AB, Kirchner P, Stürzl M, Sexton D, Palliser D, Atreya R, Siegmund B; TRR241 IBDome consortium; Neurath MF, Zundler S. Differential effects of ontamalimab versus vedolizumab on immune cell trafficking in intestinal inflammation and IBD. J Crohns Colitis. 2023 May 19:jjad088.
Reviewer 2 Report
Review for the manuscript “Effects of golimumab and ustekinumab on circulating dendritic cell migratory capacity in inflammatory bowel disease”
Dear Editor, this is an exciting manuscript. However, I have some suggestions before it can be accepted for publication.
Overall comments: This is a study where authors intended to evaluate the expression of several homing markers and the migratory capacity of circulating DC subsets toward intestinal chemo-attractants and assess the effect elicited by golimumab and ustekinumab on these characteristics in IBD compared with healthy controls.
ABSTRACT
In lines 11-25 we can read “…Inflammatory bowel disease (IBD) is a chronic disorder with a not well understood etiology, which includes ulcerative colitis (UC) and Crohn’s disease (CD). Both diseases are characterized by an uncontrolled intestinal immune response that generates tissue inflammation. Dendritic cells (DC) are antigen-presenting cells that play a key role in tolerance maintenance in the gastrointestinal mucosa. Previous studies indicated that DC recruitment by the intestinal mucosa is more prominent in IBD patients; nevertheless, the specific mechanism governing this migration is currently unknown. This study evaluated the expression of several homing markers and the migratory capacity of circulating DC subsets toward intestinal chemo-attractants, and assessed the effect elicited by golimumab and ustekinumab on these characteristics in IBD compared with healthy controls. Our results revealed that type 2 conventional DC (cDC2) had a more prominent migratory profile than other DC subsets. In addition, golimumab and ustekinumab modified the migratory profile of cDC1 differentially in CD and UC. Although in this cohort of patients CCL2 did not exert a chemoattractant effect on DC subsets, when these cells were treated with golimumab and ustekinumab their migratory capacity toward CCL2 and CCL25 was increased while that toward Mad-Cam1 was decreased. Our results indicate that cDC2 present a more migratory profile toward the gastrointestinal mucosa than other DC subsets, which should be further studied to develop new diagnostic biomarkers or to identify immunomodulatory targets in IBD.”
Please see that there are some mistakes regarding English spelling. Please check along with all the text.
KEYWORDS
I suggest including “ulcerative colitis and Crohn’s disease” and removing “migratory capacities”
.
INTRODUCTION
Please include other references published in 2022 and 2023. The authors can find good new possibilities in PUBMED.
Several abbreviations are used in this section without the definitions the first time it is seen. As example: CCR9, TNF-alpha, CCL25, CCR2, CCL20…
METHODS
In lines 97-98 we can read, “The study was approved by the local ethics committee at La Princesa Hospital (Madrid, Spain). All patients gave written informed consent for their participation and sample collection.” Please include the date of the approval.
RESULTS
The figures are excellent. However, they are too small, especially Figures 4 and 5. In this case, it is impossible to understand histogram bars.
DISCUSSION
This section is adequate. Please include the limitations of this study.
Can authors find references published in 2023?
CONCLUSION
This section is adequately described.
REFERENCES
As pointed out above, include newer references in the text.
FINAL COMMENTS:
I suggest that the authors double-check punctuation and grammar.
Minor corrections are necessary.
Author Response
Reviewer #2
Review for the manuscript “Effects of golimumab and ustekinumab on circulating dendritic cell migratory capacity in inflammatory bowel disease”
Dear Editor, this is an exciting manuscript. However, I have some suggestions before it can be accepted for publication.
Overall comments: This is a study where authors intended to evaluate the expression of several homing markers and the migratory capacity of circulating DC subsets toward intestinal chemo-attractants and assess the effect elicited by golimumab and ustekinumab on these characteristics in IBD compared with healthy controls.
ABSTRACT
In lines 11-25 we can read “…Inflammatory bowel disease (IBD) is a chronic disorder with a not well understood etiology, which includes ulcerative colitis (UC) and Crohn’s disease (CD). Both diseases are characterized by an uncontrolled intestinal immune response that generates tissue inflammation. Dendritic cells (DC) are antigen-presenting cells that play a key role in tolerance maintenance in the gastrointestinal mucosa. Previous studies indicated that DC recruitment by the intestinal mucosa is more prominent in IBD patients; nevertheless, the specific mechanism governing this migration is currently unknown. This study evaluated the expression of several homing markers and the migratory capacity of circulating DC subsets toward intestinal chemo-attractants, and assessed the effect elicited by golimumab and ustekinumab on these characteristics in IBD compared with healthy controls. Our results revealed that type 2 conventional DC (cDC2) had a more prominent migratory profile than other DC subsets. In addition, golimumab and ustekinumab modified the migratory profile of cDC1 differentially in CD and UC. Although in this cohort of patients CCL2 did not exert a chemoattractant effect on DC subsets, when these cells were treated with golimumab and ustekinumab their migratory capacity toward CCL2 and CCL25 was increased while that toward Mad-Cam1 was decreased. Our results indicate that cDC2 present a more migratory profile toward the gastrointestinal mucosa than other DC subsets, which should be further studied to develop new diagnostic biomarkers or to identify immunomodulatory targets in IBD.”
Please see that there are some mistakes regarding English spelling. Please check along with all the text.
We thank to the reviewer for his accurate comment, we rewrite the abstract.
Inflammatory Bowel Disease (IBD) is a chronic condition which include ulcerative colitis (UC) and Crohn's disease (CD), the origins of which are not yet fully understood. Both conditions involve an exacerbated immune response in the intestinal tract, leading to tissue inflammation. Dendritic cells (DCs) are antigen-presenting cells crucial for maintaining tolerance in the gastrointestinal mucosa. Previous research has indicated that DC recruitment to the intestinal mucosa is more pronounced in individuals with IBD, but the specific mechanisms governing this migration remain unclear. This study aimed to assess the expression of various homing markers and the migratory abilities of circulating DC subsets in response to intestinal chemotactic signals. Additionally, the study examined how golimumab and ustekinumab impact these characteristics in individuals with IBD compared to healthy controls. The findings revealed that a particular subset of DCs known as type 2 conventional DCs (cDC2) displayed a more pronounced migratory profile compared to other DC subsets.Furthermore, the study observed that golimumab and ustekinumab had varying effects on the migratory profile of cDC1 in individuals with CD and UC. While CCL2 did not exert a chemoattractant effect on DC subsets in this patient cohort, treatment with golimumab and ustekinumab enhanced their migratory capacity towards CCL2 and CCL25 while reducing their migration towards MadCam1.In conclusion, the study highlights that cDC2 exhibit a heightened migratory profile towards the gastrointestinal mucosa compared to other DC subsets. This finding could be explored further for the development of new diagnostic biomarkers or the identification of potential immunomodulatory targets in the context of IBD.
KEYWORDS
I suggest including “ulcerative colitis and Crohn’s disease” and removing “migratory capacities”
We modified the key words.
INTRODUCTION
Please include other references published in 2022 and 2023. The authors can find good new possibilities in PUBMED.
We thank to the reviewer for his comment, we include the following references
Agrawal, M.; Jess, T. Implications of the Changing Epidemiology of Inflammatory Bowel Disease in a Changing World. United Eur. Gastroenterol. J. 2022, 10, 1113–1120
Henry CM, Castellanos CA, Reis e Sousa C. DNGR-1-mediated cross-presentation of dead cell-associated antigens. Semin Immunol. 2023 Mar;66:101726.
Sun D, Li C, Chen S, Zhang X. Emerging Role of Dendritic Cell Intervention in the Treatment of Inflammatory Bowel Disease. Biomed Res Int. 2022 Oct 10;2022:7025634.
Several abbreviations are used in this section without the definitions the first time it is seen. As example: CCR9, TNF-alpha, CCL25, CCR2, CCL20…
We thank to the reviewer for his accurate comment, we include the definitions along to the manuscript.
Line 59: groups: plasmacytoid DCs (pDCs), identified by the presence of CD123 and Breast Cancer Type 2 susceptibility protein (BDCA2) markers (CD123+ BDCA2+), and conventional DCs
Line 62: subdivisions into two types: type 1 (cDC1) characterized by the presence of CD141 and C-X-C Motif Chemokine Receptor 1(CXCR1) markers (CD141+ CXCR1+), and type 2 (cDC2)
Line 64: identified by CD1c and signal regulatory protein α (SIRPα) markers (CD1c+ SIRPα+).Circulating cDC1 are specialized in a process known as cross-presentation
Line 72: previously demonstrated that CD patients show a higher number of interleukin (IL)-23
Line 73: and tumor necrosis factor alpha (TNFα) producing DC involved in disease pathogenesis
Line 80: Additionally, the chemokine receptor 9 (CCR9) is involved in cDC migration to the GI
Line 81: CCR9 binds to its ligand C-C Motif Chemokine Ligand 25 (CCL25), which is secreted in
Line 85: colon. It interacts with C-C Motif Chemokine Ligand 20 (CCL20), which is secreted by
METHODS
In lines 97-98 we can read, “The study was approved by the local ethics committee at La Princesa Hospital (Madrid, Spain). All patients gave written informed consent for their participation and sample collection.” Please include the date of the approval.
We thank to the reviewer for his comment , we include the ethical protocol number (JPG-VED-2016-01) in the manuscript.
Line 129: malignancy diseases. This study (JPG-VED-2016-01) received ethical approval
RESULTS
The figures are excellent. However, they are too small, especially Figures 4 and 5. In this case, it is impossible to understand histogram bars.
We would like to express our gratitude to the reviewer for their valuable comment. As suggested, the sizes of the figures have been increased.